# Evolution of hospitalized patient characteristics through the first three COVID-19 waves in Paris area using machine learning analysis

Camille Jung[1], Jean-Baptiste Excoffier[2]*, Mathilde Raphaël-Rousseau[3], Noémie Salaün-Penquer[2], Matthieu Ortala[2], Christos Chouaid[4,5]

**1** Clinical Research Center, CHI Créteil, Créteil, France, **2** Kaduceo, Toulouse, France, **3** Department of medical information, CHI Créteil, Créteil, France, **4** Department of pneumology, CHI Créteil, Créteil, France, **5** Inserm U955, UPEC, IMRB, Créteil, France

* jeanbaptiste.excoffier@kaduceo.com

**Data Availability Statement:** Data cannot be shared publicly because they are PMSI data. Even though they concern deidentified data, their use is

## Abstract

Characteristics of patients at risk of developing severe forms of COVID-19 disease have been widely described, but very few studies describe their evolution through the following waves. Data was collected retrospectively from a prospectively maintained database from a University Hospital in Paris area, over a year corresponding to the first three waves of COVID-19 in France. Evolution of patient characteristics between non-severe and severe cases through the waves was analyzed with a classical multivariate logistic regression along with a complementary Machine-Learning-based analysis using explainability methods. On 1076 hospitalized patients, severe forms concerned 29% (123/429), 31% (66/214) and 18% (79/433) of each wave. Risk factors of the first wave included old age ($\geq$ 70 years), male gender, diabetes and obesity while cardiovascular issues appeared to be a protective factor. Influence of age, gender and comorbidities on the occurrence of severe COVID-19 was less marked in the 3rd wave compared to the first 2, and the interactions between age and comorbidities less important. Typology of hospitalized patients with severe forms evolved rapidly through the waves. This evolution may be due to the changes of hospital practices and the early vaccination campaign targeting the people at high risk such as elderly and patients with comorbidities.

## Introduction

The world has been facing a pandemic of COVID-19 caused by the SARS-Cov-2 virus since March 2020. COVID-19 causes a respiratory tract infection of varying severity. It has been estimated that approximately 20% of hospitalized patients develop a severe respiratory infection requiring admission to the Intensive Care Units (ICU) or leading to death [1]. During an epidemic wave, the virus spreads very quickly in the population and the hospitals are overloaded. In particular, ICU for patients with organ failure are insufficient in many countries affected by

under strict restrictions imposed by the 'National Commission on Informatics and Liberty' (CNIL acronym in French). Here is the official website in English of the CNIL (https://www.cnil.fr/en/official-texts), with the specific CNIL methodology for data usage in French (https://www.cnil.fr/fr/declaration/mr-005-etudes-necessitant-lacces-aux-donnees-du-pmsi-etou-des-rpu-par-les-etablissements). Still, one can contact the Data Protection Officer (DPO) of the Creteil hospital for further information about data. The DPO is currently Rebecca Dadi, with the email address being dpo@chicreteil.fr.

**Funding:** The authors received no specific funding for this work.

**Competing interests:** The authors have declared that no competing interests exist.

**Abbreviations:** CHIC, Centre Hospitalier Intercommunal Créteil; CNIL, Commission Nationale de l'Informatique et des Libertés; GDPR, General Data Protection Regulation; IBD, Inflammatory Bowel Disease; ICD-10, International Classification of Diseases standard v10; ICU, Intensive Care Unit; ML, Machine Learning; PMSI, Programme de médicalisation des systèmes d'information; SHAP, SHapley Additive exPlanation; XAI, eXplainable Artificial Intelligence.

an epidemic wave [2]. Many hospitals or health agencies have developed surveillance systems to predict the intensity of epidemic waves in order to adapt the necessary medical resources, either human or material [3, 4].

Based on these surveillance systems, several risk factors for developing a severe form of COVID-19 have been identified: age, cardiovascular comorbidities, history of cancer, overweight and several clinical predicting score of worse outcomes have been developed [5, 6]. However, most studies focused on a restrained time period [5–8], and when they took into account a longer time period in order to compare the different COVID-19 waves they mainly focused on a global view of the pandemic evolution, such as numbers of infected patients, hospital admissions or death incidences [9], or were restricted to very basic medical information [10, 11]. Therefore, the evolution and interaction of the risk factors through the different waves is not clearly described. Understanding the role and interactions of risk factors associated to severe forms of COVID-19 as well as the variation in their significance during different epidemic waves are important elements in adapting the medical resources needed to manage patients in the hospital. Multivariate logistic regression is the classical method used to analyze risk factors. It provides a global vision of each feature effects, but it also suffers from several difficulties, notably concerning the non-linearities and interactions of some features. It is mostly fastidious with such type of analysis to construct a model that truly captures the effect that each feature has on an individual (a patient in our context), since a patient can suffer from more or less strong interactions between his characteristics, and in a non-linear way. While this indicates that differences between individuals are difficult to account for properly, this problem can be tackled by using state-of-the-art Machine Learning techniques that deal with non-linearity and interaction more easily, especially models based on an ensemble of decision trees, as done in [12] which benchmarked Logistic Regression versus Random Forest or [13] that highlighted the superior performances of a Boosted Tree Ensemble in a medical survival analysis context.

Then Explainability based methods, which are part of a subdomain of the Machine Learning field called Explainable Artificial Intelligence [14], also provide a way to measure the effect of each feature for each patient individually [15]. This is particularly useful for the purpose of identifying trends between variables and pointing out sub-groups of patients with common effects [16].

France is one of the most affected countries in Europe [9] and had to face 3 successive waves of COVID-19 patients, which led to 3 periods of population lockdowns. These 3 waves of COVID-19 varied in characteristics and intensity [17] in a context where vaccination against SARS-Cov-2 started in January 2021, initially targeting healthworkers and people aged 75 and older, before being progressively generalized to the adult population in June 2021 [18]. Additionally, the third wave was characterized by the emergence of the Alpha variant of concern (B.1.1.7), which became predominant from February to June 2021 [19]. These parameters could have changed the profiles of patients at risk to develop severe COVID-19. The aim of this study was to better understand the influence of age and comorbidities during the different epidemic waves. To do this, clinical risk factors associated with severe COVID-19 outcome during the 3 first waves of epidemic in a university hospital in Paris area were compared using machine learning methods.

## Patients and methods

### Study population and design

In this prospective study, all patients hospitalized because of COVID-19 at the Centre Hospitalier Intercommunal de Créteil (CHIC) with an admission stay between the 11/03/2020 and the 01/06/2021 were included. The CHIC is a 500-bed general hospital with numerous care

services and notably adult and pediatric emergency departments, a medical ICU, medicine, oncology, pneumology, pediatrics, obstetrics departments, etc.

All hospitalizations were primarily due to COVID-19, meaning that patients presented clear signs of COVID-19 infection which lead to their hospital stay or their transfer to a unit specifically dedicated to COVID-19 if they were already hospitalized. Thus all included patients had a positive Nucleic Acid Amplification Test (NAAT) or a chest tomography scan showing findings suggestive of COVID-19 disease (multiple bilateral ground glass opacities, often rounded in morphology, with peripheral and lower lung distribution) which is in accordance with the World Health Organization definition of COVID-19 [20]. No patient positive to COVID-19 but asymptomatic was included, since their hospitalization is only related but not directly caused by COVID-19 problems.

In order to assess the evolution of patient characteristics through the pandemic, three distinct periods, called waves, were defined as follows:

- Wave 1: all patients with an admission date between the 11/03/2020 and the 13/05/2020, corresponding to the first complete lockdown in France.

- Wave 2: all patients with an admission date between the 03/10/2020 and the 30/11/2020 corresponding roughly to the second France lockdown.

- Wave 3: all patients with an admission date between the 01/03/2021 and the 01/06/2021 corresponding to the third France lockdown.

The period between the first and second waves was not included since it corresponds to summer during which few patients were admitted due to COVID-19 related reasons, as well as the period between the second and third wave since it does not correspond to a new rise of cases but rather a long stagnation.

## Data collection

Data collected consists in extractions of COVID-19 hospitalized cases from the *Programme de médicalisation des systèmes d'information*, abbreviated as *PMSI*, which contains standardized medical information about hospitalizations, such as the diagnoses under the *International Classification of Diseases standard v10* (*ICD-10*). The data was entered on an ongoing basis by medical information technicians from the electronic medical record and checked by the physician in charge of the medical information (MR). Retained explanatory variables about each patient were age, gender and 18 comorbidities that were selected based on French national recommendations about people eligible for early vaccination [21, 22]. Table 1 indicates searched terms for the construction of each comorbidity.

Moreover a patient was considered to be a severe case, which is the binary target variable, if one of the following conditions was fulfilled.

- *J80** (which corresponds to Adult respiratory distress syndrome) or *R4028* (Unspecified coma) is identified in the ICD-10 diagnoses.

- Transfer to Intensive Care Unit.

- Death.

## Statistical tests and multivariate analysis

Statistical tests were performed on each wave separately in order to see the actual differences between non-severe and severe patients. Student test was used for quantitative variable (age),

**Table 1. Searched terms for binary comorbidity indicator in ICD-10.**

| Comorbidity | Searched terms |
|---|---|
| Cancer | Tumeur maligne |
| Embolism | I260, I269 |
| Cardiovascular | Hypertension, I255 |
| Cirrhosis | Cirrhose |
| Sickle Cell | D57 |
| IBD (Inflammatory bowel disease) | K55, K51, Crohn, Rectocolite |
| Mental Retardation | Retard mental |
| Cognitive Impairment | Demence, Alzheimer |
| Diabetes | E10, E11 |
| Overweight or Obesity | Surpoids, E6603, Obesite, E6604, E6605, E6606, E6609 |
| Pregnancy | Z35, Z37 |
| Trisomy | Q9 |
| Heart Failure | I5 |
| Dementia | F03 |
| Psychiatric Disorders | Troubles psychiatriques |
| Pulmonary problems | J4, J84 |
| Organ Transplant | Z94 |
| Stroke Sequelae | G81, R470 |

Searched terms corresponded to ICD-10 diagnosis id and French expression found in diagnosis description.

while the Chi-square test was applied for qualitative variables (gender, pregnancy and comorbidities, which all happen to be binary features).

Logistic regression was used to perform a classical multivariate analysis. Some comorbidities were excluded from this analysis since their low numbers prevent the computation of all p-values because of the Hessian matrix being non-invertible in the optimization process. There was no interaction term in the logistic regression analysis since interaction effects will be detailed thanks to the Machine Learning based analysis (see next subsection), but so as to fully take into account the potential non-linearity of age, which was the single continue feature, a spline based transformation of age was performed as described in [23].

## Machine learning based analysis

Another technique to analyze data is to use Machine Learning models that easier and better capture non-linearities and interactions than more traditional methods. This is the case for tree ensemble models [12, 13], and notably boosted tree ensemble in the context of COVID-19 aggravation score [24, 25]. Nevertheless, as these Machine Learning based models are more complex to interpret, explainability techniques (or *eXplainable Artificial Intelligence* abbreviated as *XAI* [14]) are applied in order to get further and more detailed insights from feature effects and interactions.

First, for each wave separately, a model was trained to distinguish between severe and non-severe cases based on all the features listed previously (age, gender and the 18 comorbidities). More precisely, to avoid biasing our results with overfitting, a K-fold cross-validation was performed with K = 5. Data for a particular wave was split into 5 distinct groups, with five steps consisting in a model learning on four groups (also called training set) then predicting only on the last group (named test set). Moreover, hyperparameter search using another 5-fold cross-

validation on the training set (technique also known as inner cross-validation) was done to optimize model performances, thus enhancing the quality of the final results.

The Machine Learning model used in our study was an ensemble of boosted trees that combines decision trees with boosting. Decision trees easily capture non-linearities and interaction effects while boosting is a technique that aggregates several submodels (here decision trees) in order to improve performances [26]. The implementation used is the *eXtreme Gradient Boosting* one, abbreviated as *XGBoost* [27].

An explicability based method was applied to have a clear view of which features contribute the most to a prediction for a particular patient. Indeed, for each patient we can have the effect, also called influence, of every feature, compared to the average prediction of the dataset. The prediction in our case was the probability of being a severe case, so the influence unit was also a probability. An influence could be negative, meaning that it decreased the probability of being a severe case thus indicating that the feature value was a protecting factor for the patient, or positive, so increasing the probability of being a severe case thus indicating that the feature value was a risk factor for the patient. For a particular patient, the sum of all its influences was equal to the difference between its prediction and the average prediction. Influences varied between each individual. For example two patients could have the exact same prediction (80% of being a severe case), but for different reasons (the first because of an old age, while the second is young but has a lot of comorbidities).

Influences were computed using the *SHapley Additive exPlanations* (*SHAP*) method [15], and particularly the *TreeSHAP* method [28] applicable to decision tree based models. In order to have the strongest possible results, influences were computed in the cross- validation process, making it a three step process: learning, prediction and influence computation.

Several visualizations were then presented to get a view of the results at different scales. First a global view was presented through a *Global Importance graph* which indicated the most important features in the model decision making. Though Global Importance graph gave a useful first glance at the feature effect, it did not link the feature initial value (for example a young age, or the presence of pulmonary problems) with its influence (which can be negative or positive). Thus it did not give any indication about the effect direction. *Distribution graph* was presented so as to get first insights about the effect direction of every feature. *Univariate graph* offered a more detailed view of a single feature effect. It is particularly useful when the feature is quantitative, as it was the case with patient age in this study. Finally, *Bivariate graph* helped visualize interaction effects of two variables.

### Ethics

This study was conducted according to the guidelines of the Declaration of Helsinki and is compliant with the GDPR (*General Data Protection Regulation*) rules and CNIL (*Commission Nationale de l'Informatique et des Libertés*) reference methodology. Analysis were performed using anonymized data.

It was approved by the Institutional Ethics Committee of Créteil hospital (*Comité d'Ethique Local du Centre Hospitalier Intercommunal de Créteil*) at the date of 24/08/2021 (approval number 2021-08-01). The use of PMSI data for research purposes does not require an individual consent. However, patients are informed on this kind of research on the hospital's website.

## Results

### Population

During the study periods, 1076 patients were hospitalized in the CHIC for COVID-19. The number of patients, number of deceases and the number and proportion of non-severe and

**Table 2. Numbers of patients and results of statistical differences between non-severe and severe cases per waves.**

| | Wave 1 | | | Wave 2 | | | Wave 3 | | | Total | | |
|---|---|---|---|---|---|---|---|---|---|---|---|---|
| | Non severe | Severe | p-value | Non severe | Severe | p-value | Non severe | Severe | p-value | Non severe | Severe | p-value |
| Nb patients | 306 (71.33) | 123 (28.67) | | 148 (69.16) | 66 (30.84) | | 354 (81.76) | 79 (18.24) | | 808 (75.09) | 268 (24.91) | |
| Nb deceases | 0 (0.0) | 70 (16.32) | | 0 (0.0) | 42 (19.63) | | 0 (0.0) | 28 (6.47) | | 0 (0.0) | 140 (13.01) | |
| Age | 59.29 (±24.44) | 71.12 (±17.2) | <0.01 ** | 60.9 (±23.58) | 74.47 (±17.55) | <0.01 ** | 56.46 (±21.57) | 62.08 (±18.73) | <0.05 * | 58.34 (±23.1) | 69.28 (±18.33) | <0.01 ** |
| Gender (women) | 151 (49.35) | 47 (38.21) | <0.05 * | 69 (46.62) | 24 (36.36) | 0.212 | 182 (51.41) | 35 (44.3) | 0.309 | 402 (49.75) | 106 (39.55) | <0.01 ** |
| Cancer | 19 (6.21) | 12 (9.76) | 0.281 | 7 (4.73) | 10 (15.15) | <0.05 * | 12 (3.39) | 5 (6.33) | 0.37 | 38 (4.7) | 27 (10.07) | <0.01 ** |
| Diabetes | 46 (15.03) | 38 (30.89) | <0.01 ** | 21 (14.19) | 17 (25.76) | 0.064 | 42 (11.86) | 13 (16.46) | 0.357 | 109 (13.49) | 68 (25.37) | <0.01 ** |
| Embolism | 7 (2.29) | 3 (2.44) | 0.795 | 1 (0.68) | 0 (0.0) | 0.678 | 11 (3.11) | 3 (3.8) | 0.97 | 19 (2.35) | 6 (2.24) | 0.898 |
| Overweight or Obesity | 50 (16.34) | 28 (22.76) | 0.155 | 21 (14.19) | 8 (12.12) | 0.848 | 43 (12.15) | 25 (31.65) | <0.01 ** | 114 (14.11) | 61 (22.76) | <0.01 ** |
| Cardiovascular | 102 (33.33) | 41 (33.33) | 1.0 | 35 (23.65) | 21 (31.82) | 0.277 | 59 (16.67) | 31 (39.24) | <0.01 ** | 196 (24.26) | 93 (34.7) | <0.01 ** |
| Cirrhosis | 0 (0.0) | 1 (0.81) | 0.637 | 0 (0.0) | 0 (0.0) | | 1 (0.28) | 0 (0.0) | 0.41 | 1 (0.12) | 1 (0.37) | 0.998 |
| Sickle Cell | 1 (0.33) | 0 (0.0) | 0.637 | 1 (0.68) | 0 (0.0) | 0.678 | 1 (0.28) | 0 (0.0) | 0.41 | 3 (0.37) | 0 (0.0) | 0.741 |
| IBD | 2 (0.65) | 0 (0.0) | 0.908 | 0 (0.0) | 0 (0.0) | | 1 (0.28) | 0 (0.0) | 0.41 | 3 (0.37) | 0 (0.0) | 0.741 |
| Mental Retardation | 3 (0.98) | 4 (3.25) | 0.208 | 0 (0.0) | 0 (0.0) | | 2 (0.56) | 0 (0.0) | 0.804 | 5 (0.62) | 4 (1.49) | 0.33 |
| Cognitive Impairment | 16 (5.23) | 8 (6.5) | 0.774 | 3 (2.03) | 3 (4.55) | 0.56 | 5 (1.41) | 3 (3.8) | 0.336 | 24 (2.97) | 14 (5.22) | 0.123 |
| Pregnancy | 23 (7.52) | 1 (0.81) | <0.05 * | 10 (6.76) | 3 (4.55) | 0.752 | 26 (7.34) | 11 (13.92) | 0.095 | 59 (7.3) | 15 (5.6) | 0.414 |
| Trisomy | 0 (0.0) | 1 (0.81) | 0.637 | 0 (0.0) | 0 (0.0) | | 1 (0.28) | 0 (0.0) | 0.41 | 1 (0.12) | 1 (0.37) | 0.998 |
| Heart Failure | 11 (3.59) | 6 (4.88) | 0.732 | 1 (0.68) | 4 (6.06) | 0.055 | 10 (2.82) | 4 (5.06) | 0.506 | 22 (2.72) | 14 (5.22) | 0.076 |
| Dementia | 0 (0.0) | 0 (0.0) | | 0 (0.0) | 0 (0.0) | | 0 (0.0) | 0 (0.0) | | 0 (0.0) | 0 (0.0) | |
| Psychiatric Disorders | 0 (0.0) | 0 (0.0) | | 0 (0.0) | 0 (0.0) | | 0 (0.0) | 0 (0.0) | | 0 (0.0) | 0 (0.0) | |
| Pulmonary problems | 33 (10.78) | 14 (11.38) | 0.993 | 9 (6.08) | 6 (9.09) | 0.612 | 25 (7.06) | 11 (13.92) | 0.076 | 67 (8.29) | 31 (11.57) | 0.136 |
| Organ Transplant | 0 (0.0) | 1 (0.81) | 0.637 | 0 (0.0) | 1 (1.52) | 0.678 | 2 (0.56) | 0 (0.0) | 0.804 | 2 (0.25) | 2 (0.75) | 0.56 |
| Stroke Sequelae | 5 (1.63) | 0 (0.0) | 0.353 | 2 (1.35) | 0 (0.0) | 0.857 | 3 (0.85) | 0 (0.0) | 0.943 | 10 (1.24) | 0 (0.0) | 0.144 |

Results are presented with mean and standard deviation for age which is a quantitative feature, and numbers and proportion for the rest since they all are binary qualitative features. A blank space for p-value means that there was no patient with the particular comorbidity in the wave, so that the Chi-square test could not be performed.

For visual help for p-value, a single star (*) denotes a p-value strictly inferior than 0.05 and two stars (**) denotes a p-value strictly inferior than 0.01.

severe cases per wave is given by Table 2. The first two waves had similar rates of severe cases and deceases, while the third wave had the lowest proportion of severe cases and deceases.

Patient age was significantly different between non-severe and severe cases for all waves, even if this was less significant for the third wave than the previous ones (Table 2). Also *Gender* is only significant for the first wave, as for *Diabetes* and *Pregnancy*. On the contrary, *Cardiovascular* problems and *Overweight or Obesity* are significantly different only for the third wave. *Cancer* is the only significant comorbidity in the second wave.

## Multivariate logistic regression analysis

A feature was included in this analysis if it came out as significant in at least one wave in the previous analysis or if at least 10 patients from the same wave had the comorbidity (Table 2). Therefore, Multivariate Logistic Regression was performed using the following explanatory variables: *Age, Gender, Cancer, Diabetes, Pulmonary problems, Embolism, Overweight or Obesity, Cardiovascular, Cognitive Impairment, Pregnancy, Heart Failure.*

First of all, in order to fully capture the non-linearity effect of age, which was the single quantitative variable in this study, a transformation using cubic splines was applied for each wave. The resulted transformations are given by Fig 1. The probability of being a severe case increased with age for all waves. The only exception was the first wave which was not strictly monotonic since the risk effect of age peaked at about 80 years, while older patients had a little smaller probability of being severe cases than 80 year old patients. It is worth noticing that the third wave had the lowest gap between younger and older ages in terms of risk effect.

Table 3 indicates the results in terms of odds-ratio and p-values for each wave, with a single logistic regression having been run for each of them. The *Age* feature is taken after the spline transformation. *Age* always came out as significant at an $\alpha$-level of 5%, it even came out for the two first waves at a lower 1% $\alpha$-level. *Gender* was significant for the two last waves, whereas *Overweight or Obesity* and *Cardiovascular* were significant for both the first and third waves and *Pregnancy* came out to be significant for the last two waves.

## Machine learning analysis

Table 4 displays performance metrics for XGBoost model for each wave, including accuracy and area under the curve of receiver operating characteristic (AUC ROC Score).

After learning models on each wave separately and computing influences for each patient with the TreeSHAP method, a first way to apprehend what models had learnt from the data is to look at the most important features in their decision making. Fig 2 shows the *Global Importance graph* for each feature and wave. This gives an interesting first glance to easily capture feature importance through the waves.

*Age* was the predominant variable for every wave, especially the second one. Gender was also an important feature for all three waves. *Overweight or Obesity* and *Cardiovascular* were influential only for the first and third waves. Diabetes was important only for the first wave, but also in a weaker manner for the last one. On the contrary, *Pulmonary problems* and *Pregnancy* were important for the third wave only. This is mainly coherent with the multivariate analysis through logistic regression.

Though Global Importance values gave a useful first glance at the feature effect, it did not link the feature initial value (for example a young age, or the presence of pulmonary problems) with its influence (which can be either negative, null or positive). Hence it did not give any indication about the direction of the effects. Distribution graph for each wave are given in Fig 3, which is a first way to get first insights about the effect direction of each feature per wave.

The risk of being a severe case increased with *Age*, since for all waves there were more young patients (represented by blue points) that were associated with negative influences, while older patients (indicated by redder points) were mostly associated with positive influences. Nevertheless, this distinction between young and old ages was particularly true for the first and second waves, while it was more noisy and heterogeneous for the third wave. As for the *Gender* feature it came out that men were at higher risk of being severe cases in all waves and especially the first one. For almost all comorbidities the presence was a risk factor. It was the case for *Diabetes* and *Overweight or Obesity* for the first wave, and *Overweight or Obesity, Cardiovascular, Pulmonary problems* and *Pregnancy* for the third wave. The two exceptions

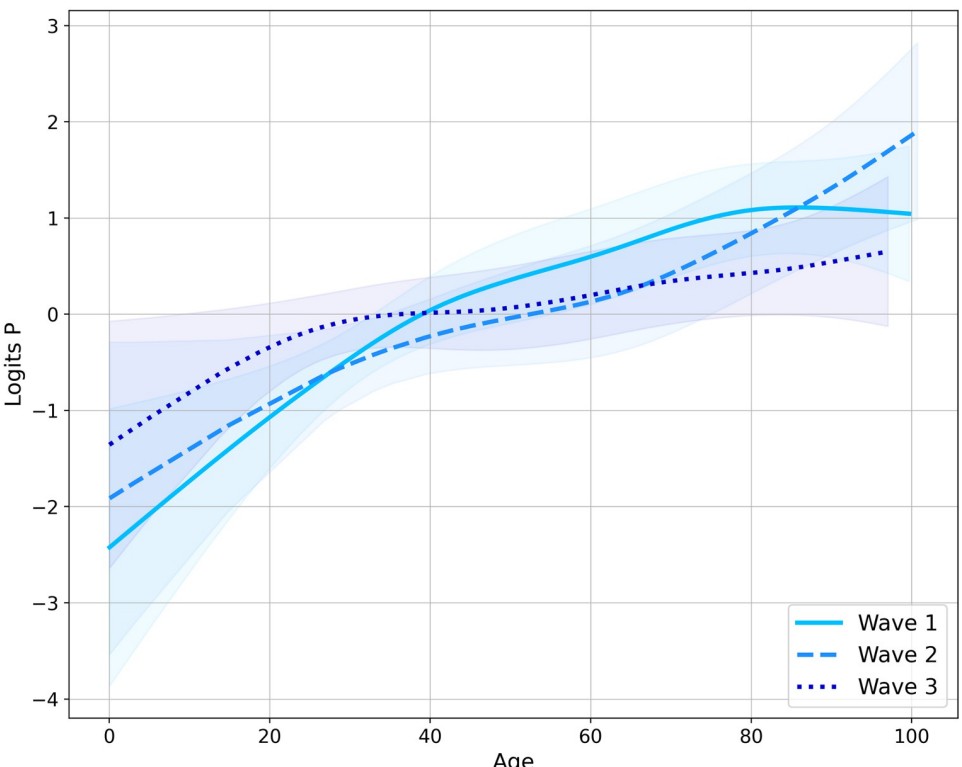

**Fig 1. Spline transformation of *Age* for each wave.** Shaded areas represent confidence intervals.

were *Cardiovascular* and *Diabetes* that were considered as protective factor especially for the *Cardiovascular* feature, respectively for the first and last wave. No comorbidity came out as significant in the second wave.

An Univariate visualization was also made to get a more detailed view of feature effect. It is particularly useful when the feature is quantitative, as it is the case with patient age in this

**Table 3. Results of multivariate logistic regression analysis.**

|  | Wave 1 | | Wave 2 | | Wave 3 | |
|---|---|---|---|---|---|---|
|  | Odds Ratio (95% CI) | p-value | Odds Ratio (95% CI) | p-value | Odds Ratio (95% CI) | p-value |
| constant | 0.2 (0.11-0.35) | <0.01 ** | 0.2 (0.1-0.39) | <0.01 ** | 0.08 (0.04-0.15) | <0.01 ** |
| Age | 3.37 (1.83-6.21) | <0.01 ** | 3.62 (1.91-6.85) | <0.01 ** | 9.95 (1.64-60.45) | <0.05 * |
| Gender | 0.66 (0.41-1.05) | 0.077 | 0.47 (0.23-0.94) | <0.05 * | 0.5 (0.28-0.92) | <0.05 * |
| Cancer | 1.39 (0.62-3.14) | 0.424 | 4.67 (1.56-13.96) | <0.01 ** | 1.49 (0.43-5.11) | 0.529 |
| Diabetes | 2.12 (1.24-3.62) | <0.01 ** | 2.22 (0.96-5.1) | 0.061 | 0.76 (0.35-1.67) | 0.492 |
| Pulmonary problems | 0.93 (0.46-1.91) | 0.852 | 1.46 (0.43-4.9) | 0.541 | 2.42 (1.04-5.62) | <0.05 * |
| Embolism | 1.08 (0.23-4.97) | 0.922 | 0.0 (0.0-76.1) | 0.969 | 1.57 (0.39-6.35) | 0.531 |
| Overweight or Obesity | 1.86 (1.03-3.33) | <0.05 * | 1.31 (0.49-3.51) | 0.594 | 4.38 (2.24-8.56) | <0.01 ** |
| Cardiovascular | 0.55 (0.33-0.91) | <0.05 * | 0.8 (0.37-1.7) | 0.559 | 2.61 (1.36-5.01) | <0.01 ** |
| Cognitive Impairment | 0.82 (0.33-2.08) | 0.683 | 1.8 (0.29-10.93) | 0.525 | 1.84 (0.34-10.01) | 0.482 |
| Pregnancy | 0.5 (0.06-4.24) | 0.522 | 5.43 (1.05-28.03) | <0.05 * | 9.61 (3.23-28.61) | <0.01 ** |
| Heart Failure | 1.21 (0.41-3.57) | 0.73 | 5.29 (0.51-54.76) | 0.163 | 0.89 (0.23-3.4) | 0.866 |

For visual help for p-value, a single star (*) denotes a p-value strictly inferior than 0.05 and two stars (**) denotes a p-value strictly inferior than 0.01.

**Table 4. XGBoost performance metrics for each wave.**

| Metrics | Wave 1 | Wave 2 | Wave 3 |
|---|---|---|---|
| Accuracy (%) | 70.63 | 66.82 | 70.67 |
| AUC ROC Score (%) | 65.52 | 65.77 | 62.35 |

Results were obtained after a 5-fold cross-validation process, including an inner cross-validation for hyperparameter optimization.

study, as shown by Fig 4 which clearly shows that *Age* effect greatly varies between waves. The first wave had a slight increase of the risk until about the age of 65, then the risk rose suddenly to the age of 80, which was the risk peak for this wave. Then there was a little decrease of the risk for patients older than 80 years. On the contrary the second wave had a strong threshold effect of *Age*. Patients younger than the age of 70 were more or less given the same risk, then between 70 and 80 year old there was a rapid increase of the risk, and finally patients aged 80 years or more were also given a similar risk. The third wave was significantly different from the first two, since influences were more heterogeneous. Indeed, some young patients were given a positive influence (indicating an high risk of being severe cases) while some older patients on the contrary were given negative influences (low risk). Nevertheless, on average young patients were still less at risk than older ones, but in a less neat way than previous waves.

Bivariate plots indicating interaction between *Age* and four binary features are shown in Fig 5. Age was selected as it was the most important feature for all waves (Fig 2), and the four binary features (*Gender, Diabetes, Overweight or Obesity* and *Cardiovascular*) were chosen since they all were the second most important feature for at least one wave (Fig 2). There was a

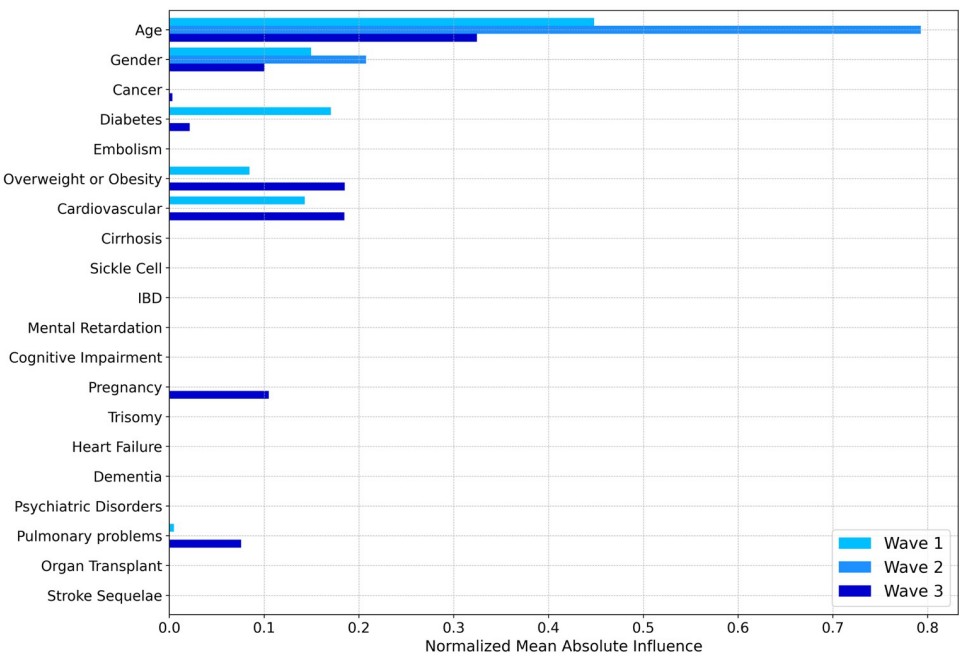

**Fig 2. *Global variable importance* for each wave.** Global Importance value indicates how important a feature in a particular wave is. As influences can be either positive or negative depending on the patient, Global Importance value for a feature and a wave is computed by averaging the absolute influence for that feature over all patients from the wave, then normalizing by the sum of all Global Importance feature values from the wave. Thus the sum of Global Importance values over a single wave is equal to 1.

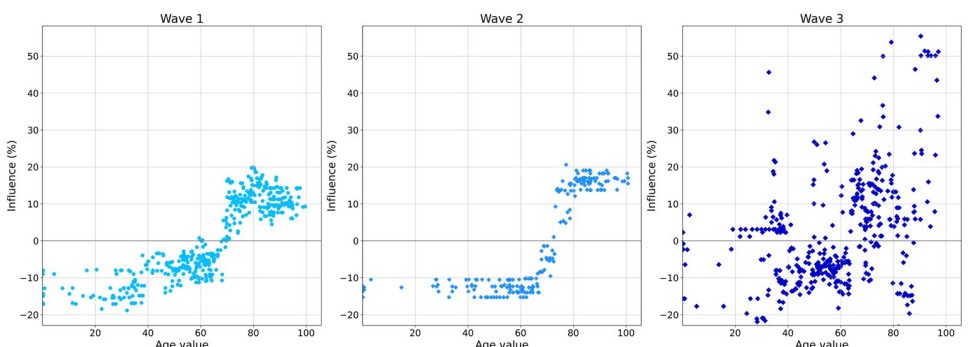

**Fig 3. *Distribution graph* of influence with respect to feature value for each feature per wave (best viewed in color).** Each dot represents a particular patient from the dataset, with feature names on the y-axis and influence values on the x-axis. A negative influence can thus be considered as a protective factor since it decreased the probability of being in a severe state, whereas a positive influence can be considered as a risk factor. The initial value is represented by the color of the point, through the colormap indicated on the far right of the figure. A low initial value (for example a young age) is indicated by a blue point, while a high initial value (an old age) would be a red one. *Gender* value is 0 for men (blue dots) and 1 for women (red dots) while Comorbidity value is 0 for the absence (blue) and 1 for the presence (red).

**Fig 4. *Univariate graph* of patient *Age* effects for each wave.** Each dot represents a patient, with age value on the x-axis and the associated influence on the y-axis.

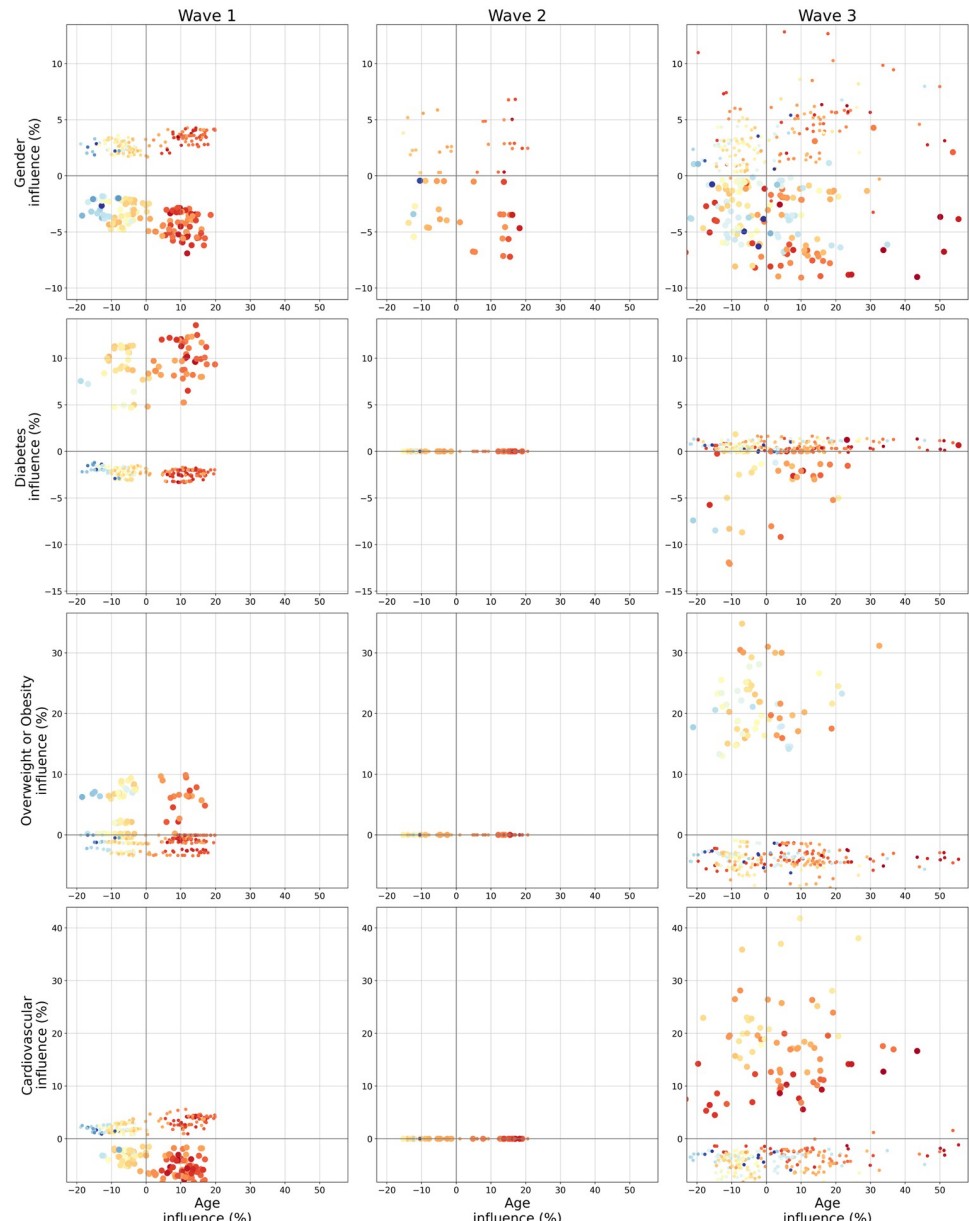

**Fig 5.** *Bivariate graph* **for** *Age* **and its interaction with** *Gender, Diabetes, Overweight or Obesity* **and** *Cardiovascular* **for each wave (best viewed in color).** For each row, *Age* influence is on the x-axis while *Gender, Diabetes, Overweight or Obesity* and *Cardiovascular* are on the y-axis respectively for the first, second, third and fourth row. Each dot represents a particular patient. Color and size of the dot give information about the two feature initial values, respectively color for the feature located on the x-axis (here age) and size for the feature on the y-axis. The colormap for the x-axis feature is the same used in Fig 3 with blue indicating a young age and red an old age. Since the y-axis features are all binary indicators, a small point indicates a 0 (men or absence of comorbidity) and a bigger dot a 1 (women or comorbidity presence).

clear interaction effect between *Age* and *Gender* for the first wave. Indeed, all men were given a higher risk than women, and the difference got larger as age increased. It does indicate that *Gender* had a stronger effect when patients were older. Nevertheless, this fact was not as clear for the second and third waves, especially the latter.

In a similar fashion, *Diabetes* effect got stronger with *Age* for the first wave. It is also noticeable that *Diabetes* had no effect for the second wave, and that it was a protective factor for the third wave even if this was only a slight effect for most patients. As for the interaction effects between *Age* and *Overweight or Obesity*, only the first and third waves saw an non-null effect of *Overweight or Obesity*, but seemingly without interaction with *Age*.

Finally concerning the interaction of *Age* and *Cardiovascular* problems the second wave once again had no comorbidity effect. But the first and third waves showed inverse effects. While the presence of *Cardiovascular* problems was a risk factor in the third wave, it was on the contrary a protective factor for the first wave. Moreover, for both waves this effect was related to *Age*, but again in an inverse fashion. While for the third wave the *Cardiovascular* effect decreased with *Age* since older patients with cardiovascular problems (indicated by red big points) were given lower influences than younger patients with cardiovascular problems (yellow and orange big points), for the first wave it got stronger with *Age*.

## Discussion

This study of over 1000 COVID-19 cases highlighted changes in the influence of clinical risk factors over the first 3 waves of the epidemic in an University hospital in Paris area thus offered a deeper understanding of the pandemic through a long term analysis of its evolution, which very few studies or research notes have done until now [10, 11].

While a higher age ($\geq 70$ years) was a risk factor to develop a severe form for all of the 3 waves, the ML analyses underlined a more heterogeneous impact of this feature during the third wave, indicating that an older age was no longer a clear risk factor as it was for the first two waves. Moreover, some risk factors appeared with time, such as pregnancy or having an underlying chronic lung disease. In a similar manner, interactions between features evolved rapidly. In the first wave age highly interacted with other variables such as gender or cardiovascular problems, while this was not the case for the following waves, with the second wave not even having any impactful comorbidity. These results are coherent with the existing literature that mainly focused on the first waves of the pandemic [7, 8, 29]. Thus, Williamson et al [30] analyzed the data from the British electronic health database on more than 17 million of individuals during the first wave of COVID-19 using multivariate Cox model. Age, gender, obesity, diabetes, cancer and immunosuppression were identified as risk factors, in accordance with the results found in this study during the first wave. The only exception concerned cardiovascular problems for the first wave that came out as a slight protective factor in this study, both in the logistic regression and ML analysis, which is in contradiction with other researches [31]. It could come from the fact that the CHIC hospital has not a large cardiology department. Thus some patients with heavy cardiovascular problems could have been redirected from the CHIC to another hospital in the first wave since it was the peak of the pandemic. Additionally, some prognosis scores for mortality in SARS-Cov-2 pneumonia have been developed [5, 6]. These scores integrated age and comorbidities risk factors but also biological, radiological or genetic factors that are not immediately or routinely available.

The changes in these clinical risk factors can be explained by a change in the exposed population as the virus spread in the population between waves 1 and 2 and by the vaccination campaign for the elderly ($\geq 75$ years) and people "at risk" for the third wave [32]. Thus, it is expected that the generalization of vaccination in a context where the virus and its recent variant forms continues to be very present in the general population will modify the profile of people severely affected by the disease. In particular, as vaccination is seemingly efficient on severe forms [33] and that most people with high risk factors (elderly, comorbidity) are vaccinated

[32], it is therefore expected that severe forms of COVID-19 will mainly affect unvaccinated people who have fewer risk factors and comorbidities.

Even though this study was monocentric, it did include data for more than a year from an important University Hospital located in a dense population area. This study included rather classical clinical information (Age, Gender and numerous comorbidities) but no biological parameters. Nevertheless, informations were extracted from a well formatted database that is mandatory for every hospital center in France (PMSI), and all these data were verified by the physician in charge of coding, in order to guarantee the reliability of the data. All features were therefore easily constructed and can be considered as reliable and also time stable, in contrast to biological variables that can heavily fluctuate since they depend on the moment the measurement is done.

Performances of the XGBoost model were rather low for all waves. This probably came from the fact that this study only included easily available features (age, gender and comorbidities) but no more complex radiological and biological variables, therefore limiting the discriminative power of models applied on these data. Despite these modest performances, the solid model building and validation process, using nested cross-validation, allowed the Machine Learning based analysis to give complementary insights and understanding about the patient typology and its evolution through the pandemic. It was particularly useful to identify subgroups with similar effects, as well as the visualization of the diffusion level of the effects. Thus, it offers an almost real-time monitoring of the evolution of risk factors, enabling a rapid adaptation and improvement of the healthcare services.

Future works could involve a study with a population from a larger area in order to evaluate the stability of the findings and measure the differences between regions. Additional features such as biological variables, including C-Reactive Protein [34] or Eosinophils [35], could also be included to get a finer comprehension but such information are not the most reliable and are more difficult to gather.

Moreover, while Machine Learning based analysis brings relevant information about individual patient protective and risk factors, as well as their degree of homogeneity or heterogeneity among the general population, it is essentially visual for now. Hence to some extent it lacks statistical and global quantitative information about each feature, as it can be done with more classical analysis methods such as logistic regression but at the cost of a rather fastidious model construction and higher complexity since it is difficult to individually compute actual effect of each feature if the logistic regression model has interaction terms and non-linear transformations. Therefore, the establishment of techniques that would provide a more quantitative view of the general behavior of the features along with statistical testing of differences based on influence values are interesting future axis of work in order to provide healthcare practitioners and decision-makers an even quicker and better understanding of the epidemic situation so they can efficiently choose the most appropriate responses and necessary adjustments.

## Conclusions

This study highlighted significant changes of COVID-19 characteristics for severe state hospitalized patients over the first 3 waves of the pandemic in France. In complement to classical analysis methods, Machine Learning-based techniques emphasized on the heterogeneity level of effects and feature interactions. This rapid evolution can be explained by changes in hospital practices as well as vaccination campaign that targeted first and foremost people most at risk. It is thus expected that severe COVID-19 forms will affect unvaccinated people with fewer risk factors and comorbidities.

## Author Contributions

**Conceptualization:** Camille Jung, Jean-Baptiste Excoffier, Noémie Salaün-Penquer, Matthieu Ortala, Christos Chouaid.

**Data curation:** Jean-Baptiste Excoffier, Mathilde Raphaël-Rousseau, Noémie Salaün-Penquer.

**Formal analysis:** Camille Jung, Matthieu Ortala.

**Investigation:** Jean-Baptiste Excoffier.

**Methodology:** Camille Jung, Jean-Baptiste Excoffier.

**Supervision:** Camille Jung, Noémie Salaün-Penquer, Matthieu Ortala, Christos Chouaid.

**Validation:** Camille Jung, Jean-Baptiste Excoffier, Mathilde Raphaël-Rousseau, Noémie Salaün-Penquer, Matthieu Ortala, Christos Chouaid.

**Visualization:** Jean-Baptiste Excoffier.

**Writing – original draft:** Camille Jung, Jean-Baptiste Excoffier.

**Writing – review & editing:** Camille Jung, Mathilde Raphaël-Rousseau, Noémie Salaün-Penquer, Matthieu Ortala, Christos Chouaid.

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
