## [Decision Letter · Decision Letter 0]

8 Nov 2021

PONE-D-21-30788Evolution of hospitalized patient characteristics through the first three COVID-19 waves in Paris area using machine learning analysisPLOS ONE

Dear Dr. Excoffier,

Thank you for submitting your manuscript to PLOS ONE. After careful consideration, we feel that it has merit but does not fully meet PLOS ONE’s publication criteria as it currently stands. Therefore, we invite you to submit a revised version of the manuscript that addresses the points raised during the review process.

We look forward to receiving your revised manuscript.

Kind regards,

Kanhaiya Singh, Ph.D

Academic Editor

PLOS ONE

Additional Editor Comments (if provided):

Although the reviewers have found this study interesting, they have recommended revision to improve the clarity of the manuscript. Please provide codes and other details of the model as recommended by Reviewer 1.

3. PLOS requires an ORCID iD for the corresponding author in Editorial Manager on papers submitted after December 6th, 2016. Please ensure that you have an ORCID iD and that it is validated in Editorial Manager. To do this, go to ‘Update my Information’ (in the upper left-hand corner of the main menu), and click on the Fetch/Validate link next to the ORCID field. This will take you to the ORCID site and allow you to create a new iD or authenticate a pre-existing iD in Editorial Manager. Please see the following video for instructions on linking an ORCID iD to your Editorial Manager account: https://www.youtube.com/watch?v=_xcclfuvtxQ. 

4. We note you have included a table to which you do not refer in the text of your manuscript. Please ensure that you refer to Table 1 in your text; if accepted, production will need this reference to link the reader to the Table.

Reviewers' comments:

Reviewer's Responses to Questions

**Comments to the Author**

1. Is the manuscript technically sound, and do the data support the conclusions?

Reviewer #1: Partly

Reviewer #2: Yes

2. Has the statistical analysis been performed appropriately and rigorously? 

Reviewer #1: I Don't Know

Reviewer #2: I Don't Know

3. Have the authors made all data underlying the findings in their manuscript fully available?

Reviewer #1: No

Reviewer #2: Yes

4. Is the manuscript presented in an intelligible fashion and written in standard English?

Reviewer #1: Yes

Reviewer #2: No

5. Review Comments to the Author

Reviewer #1: Summary:

The article provides an interesting analysis of patient characteristics hospitalized due to COVID-19 at the Centre Hospitalier Intercommunal de Cr´eteil using logistic regression and boosted decision trees. I have comments that should be addressed for publication. I would ask the authors to provide more information on the boosted decision trees. Specifically, the number of nodes and leaves should be reported (preferably visualized), the model’s prediction outcomes for each wave should be reported in a confusion matrix, and accuracy vs loss plots for the K-fold cross validation provided.

Major Issues:

1. Multiple authors work for/founded a private company, Kaduceo (https://kaduceo.com/, https://www.linkedin.com/company/kaduceo/), focused on explainable artificial intelligence for healthcare applications. Given the emphasis on the value of services provided by this company, why is this not disclosed in the Conflict of Interest section? Were any of the authors paid to perform this work?

2. The code should be provided in the supplement. Given the restrictions on the data this is necessary to verify the methodology.

3. Please provide a confusion matrix, or similar graphic, in the supplement for the prediction outcomes of the model for each wave. Accuracy vs Loss plots should be provided for the k-fold cross validation for each wave.

4. The report of cardiovascular conditions providing a protective influence on severe COVID cases appears to be in conflict with a recent report by Banerjee et al 2021 (https://doi.org/10.1093/eurjpc/zwaa155). Can the authors provide additional rationale to distinguish the findings between these studies?

5. Line 216 needs to be rephrased. “Bluer” and “redder” are visualization aids, not data descriptors. The authors obviously understand this given figure 3’s caption, but this should be reflected in the main text. Similarly, lines 262 and 263 need to be rephrased to provide a proper description rather than “big redder points” and “other big points.”

Minor Issues:

6. Image resolution must be improved for publication.

7. Figure colors should be adapted for colorblind readers (https://adasitecompliance.com/ada-color-contrast-checker/).

8. The descriptor that this is a large study should be changed (line 265). This study has value but the scope is relatively narrow compared to similar studies (https://doi.org/10.1016/j.scitotenv.2021.145650, https://doi.org/10.1016/j.scitotenv.2020.142810, https://doi.org/10.1016/j.chaos.2020.110058).

9. Please provide citations for lines 98, 105, and 122.

10. Please review for grammar and typos (lines 13, 20, 24, 58, 74, 78, 188, 193, 199, 234, 240, 277, 279, 300, 315, 327).

Reviewer #2: Ref: PONE-D-21-30788

In the present article entitled "Evolution of hospitalized patient characteristics through the first three COVID-19 waves in Paris area using machine learning analysis," Jung et al. have used machine-based learning to analyze the trends in risk factors associated with COVID-19 through peaks in the pandemic in Paris. The study design is well conceived and the results are well supported by the data.

Especially liked the fact that the authors have noted the limitations of the study in the manuscript.

However, some points need to be addressed to make the study robust for publication.

Major:

Abstract mentions.. 'Data was prospectively collected from a University Hospital in Paris

area, over..' it appears that the data was collected retrospectively from a prospectively maintained database..

Methodology: The authors should clarify the Inclusion criteria section to avoid confusion… for example:

The inclusion criteria of the study appear to be as per the accepted WHO' Case definition': [WHO reference number: WHO/2019-nCoV/Surveillance_Case_Definition/2020.2]

It would help to know what exact criterion of either WHO or national French guidelines were followed at the hospital for admission. Especially with regards to the confirmation of clinically symptomatic cases: eg. Nucleic Acid Amplification Test (NAAT)/ positive OR SARS-CoV-2 Antigen-RDT AND meeting either the probable case definition

Kindly define the 'Primary patient catchment area for the University hospital'.

Since it looks like a referral hospital, a good number of patients might have been referred at the peak of pandemic from smaller centres with Severe disease form directly to ICU, leading to a bias in the cohort analyzed in the present study. Please justify if any measures were taken into consideration during the analysis.

Minor: Please translate French to English while describing the references for the readers easily. References: 3,20,21

Line 11: 'Based on this surveillance system...' since the antecedent has multiple systems mentioned can change to '....these surveillance systems...'

line 285: genetic factors in the singular might sound more appropriate.

Also, it would help to mention the 'biological factors' with examples affecting the pneumonia progression.

Overall, a good and relevant study.

Best regards,

6. PLOS authors have the option to publish the peer review history of their article (what does this mean?). If published, this will include your full peer review and any attached files.

Reviewer #1: No

Reviewer #2: No

---

## [Author Response · Author response to Decision Letter 0]

16 Dec 2021

Dear Reviewers and Editors,

First of all, we greatly thank editors and reviewers for their careful reading, valuable

advice and positive evaluations of our work.

Here are our responses for each review.

Journal requirements

We based our manuscript on a PLOS One template for LaTex, including the

plos2015.bst style sheet for BibTex. Still, we would gladly make any changes if

needed.

Code for model construction and validation was shared on a public GitHub

repository so that anybody can check soundness and viability of the

implementation.

ORCID of the corresponding author was entered on the PLOS One editorial

manager.

Reference to Table 1 was added.

Reviewer 1

Conflict of Interest section was modified so as to specify that indeed three of the

authors are employees of Kaduceo, but the study did not receive any specific grant.

As said above, code was shared on a public GitHub repository.

Accuracy and AUC ROC Score were added for the Machine Learning model for

each wave in a new table (Table 4), and also discussed in the Discussion section.

Reported performances were rather low due to the low number of features used

(age, gender and comorbidities). Nevertheless, the solid model construction and

validation allowed to take into account the full discriminative power potential of the

features to perform a robust and useful analysis of feature effects.

Indeed our findings about the cardiovascular problem being a protective factor for

the first wave is in contradiction with other studies. As written in the Discussion

section, this could come from the fact the hospital in this study (CHI Créteil) does

not have a large cardiology department, suggesting that some patients with heavy

cardiovascular problems were redirected to other hospital centers because of the

stressful context of the first wave.

We provided a correct description of values (such as young or old age) along with

the color indicator (blue or red).

Original images (tiff format) respected the PLOS One Figure File Requirements. We

guess it is the automatic PDF generator that blurred images, but we could provide

image with a better resolution if still needed.

Indeed this study was not large compared to some other studies, but covered a

longer period of over one year. In consequence, we remove the word ‘large’ at the

beginning of the Discussion section.

We checked that the colormap used is colorblind friendly, for example using site

such

as

ColorBrewer

(https://colorbrewer2.org/#type=diverging&scheme=RdYlBu&n=10).

References were added concerning ensemble tree models and their performances.

Typos were corrected.

Reviewer 2

Indeed, data was rather collected retrospectively from a prospectively maintained

database. Abstract was modified in consequence.

We clarified inclusion criteria in the Study population and design subsection in the

Patients and Methods section. It consisted of every patient with a positive NAAT

test or a chest tomography scan showing findings suggestive of COVID-19 disease.

Despite the hospital of this study (CHI Créteil) being a large hospital center, it was

not considered as a referral hospital of its area, notably because of the proximity of

another larger center (CHU Henri-Mondor). Moreover, the CHI Créteil had a small

Intensive Care Unit to be considered a referral hospital.

French title of certain references were translated into English.

We indeed changed ‘surveillance system’ into plural form.

‘Genetics factors’ was changed into ‘genetic factors’.

In the Discussion section, we added some biological factors considered by the

literature to have a significant impact on COVID-19 condition and aggravation, such

as C-Reactive Protein (CRP) and Eosinophils. We also put appropriate references

to support this claim.

Best Regards,

The authors

---

## [Decision Letter · Decision Letter 1]

17 Jan 2022

Evolution of hospitalized patient characteristics through the first three COVID-19 waves in Paris area using machine learning analysis

PONE-D-21-30788R1

Dear Dr. Excoffier,

We’re pleased to inform you that your manuscript has been judged scientifically suitable for publication and will be formally accepted for publication once it meets all outstanding technical requirements.

Kind regards,

Kanhaiya Singh, Ph.D

Academic Editor

PLOS ONE

Additional Editor Comments (optional):

Reviewers' comments:

Reviewer's Responses to Questions

**Comments to the Author**

1. If the authors have adequately addressed your comments raised in a previous round of review and you feel that this manuscript is now acceptable for publication, you may indicate that here to bypass the “Comments to the Author” section, enter your conflict of interest statement in the “Confidential to Editor” section, and submit your "Accept" recommendation.

Reviewer #1: All comments have been addressed

Reviewer #2: All comments have been addressed

2. Is the manuscript technically sound, and do the data support the conclusions?

Reviewer #1: Yes

Reviewer #2: Yes

3. Has the statistical analysis been performed appropriately and rigorously? 

Reviewer #1: I Don't Know

Reviewer #2: I Don't Know

4. Have the authors made all data underlying the findings in their manuscript fully available?

Reviewer #1: No

Reviewer #2: Yes

5. Is the manuscript presented in an intelligible fashion and written in standard English?

Reviewer #1: Yes

Reviewer #2: Yes

6. Review Comments to the Author

Reviewer #1: (No Response)

Reviewer #2: (No Response)

7. PLOS authors have the option to publish the peer review history of their article (what does this mean?). If published, this will include your full peer review and any attached files.

Reviewer #1: No

Reviewer #2: **Yes: **Tejas Nikumbh

---

## [Editor Report · Acceptance letter]

10 Feb 2022

PONE-D-21-30788R1 

Evolution of hospitalized patient characteristics through the first three COVID-19 waves in Paris area using machine learning analysis 

Dear Dr. Excoffier:

I'm pleased to inform you that your manuscript has been deemed suitable for publication in PLOS ONE. Congratulations! Your manuscript is now with our production department. 

Kind regards, 

on behalf of

Dr. Kanhaiya Singh 

Academic Editor

PLOS ONE